# Risk Behaviors, Family Support, and Emotional Health among Adolescents during the COVID-19 Pandemic in Israel

**DOI:** 10.3390/ijerph19073850

**Published:** 2022-03-24

**Authors:** Orit Shapiro, Rachel Nissanholtz Gannot, Gizell Green, Avi Zigdon, Moti Zwilling, Ariela Giladi, Lilach Ben-Meir, Marques Adilson, Sharon Barak, Yossi Harel-Fisch, Riki Tesler

**Affiliations:** 1Department of Health System Management, Faculty of Health Science, Ariel University, Ariel 40700, Israel; oritshapiro100@gmail.com (O.S.); rachelng@ariel.ac.il (R.N.G.); aviz@ariel.ac.il (A.Z.); 2Department of Nursing, Faculty of Health Science, Ariel University, Ariel 40700, Israel; ggrin@campus.haifa.ac.il; 3Department of Economics and Business Administration, Ariel University, Ariel 40700, Israel; motizw@gmail.com; 4School of Education, Bar Ilan University, Ramat Gan 52900, Israel; arielagiladi@gmail.com (A.G.); lilach.benmeir@gmail.com (L.B.-M.); harelyossi@gmail.com (Y.H.-F.); 5CIPER, Faculdade de Motricidade Humana, Universidade de Lisboan, 1649-004 Lisbon, Portugal; adncmpt@gmail.com; 6Program in Gerontology, Faculty of Health Sciences, Ben-Gurion University of the Negev, Beer Sheeva 8410501, Israel; sharoni.baraki@gmail.com; 7Physical Education Department, Kaye Academic College of Education, Beer Sheeva 8414201, Israel; 8Department of Pediatric Rehabilitation, The Edmond and Lily Safra Children’s Hospital, The Chaim Sheba Medical Center, Ramat Gan 5290002, Israel

**Keywords:** risk behaviors, family support, adolescents, COVID-19

## Abstract

We investigated the prevalence of risk behaviors among Israeli adolescents (tobacco smoking, alcohol consumption, drug use) during the COVID-19 pandemic. Associations between different risk behaviors were examined and so was whether specific characteristics could predict risk behaviors in adolescents. The study consisted of 1020 Israeli adolescents aged 15–18. Study subjects completed an online survey between the first and second lockdowns in Israel (April 2020 to September 2020). Participants reported the frequency at which they engaged in four different risky behaviors: general risky behavior, tobacco smoking, alcohol consumption (binge drinking), and cannabis use. The most prevalent risky behavior in the sample was binge drinking (33.8%). The four measured risky behaviors were significantly correlated. Among participants who had previously engaged in a risky behavior assessed, most did not change the behavior frequency during the pandemic. All independent variables (sociodemographic characteristics, family support, and emotional, health excluding friends’ support, physical activity volume, and coronavirus restrictions) were significantly different between participants engaging vs. not engaging in risky behaviors. Our findings suggest that family support is one of the most influential factors in preventing risky behavior during the pandemic, and they emphasize the importance of family-based interventions with children and adolescents from elementary to high school.

## 1. Introduction

Risky behavior among adolescents is a recognized worldwide phenomenon [1] that is even more relevant during times of crisis, such as the end of 2019, when the coronavirus (COVID-19) started to spread. Studies indicate various effects on adolescents since the onset of the pandemic, including a significant increase in mental distress, alcohol use, screen addiction, student dropout, decreased physical activity, and reduced involvement in community activities [2,3,4,5].

A national survey conducted in 2019 showed that the prevalence of substance use among Israeli adolescents was lower than their European counterparts (HBSC Israel survey, 2019) [6]. The outbreak of the COVID-19 pandemic in Israel led to the implementation of strict distancing measures that included the complete closure of schools and the transition to e-learning, movement and travel restriction, and a national curfew [7]. The pandemic and the confinement measures have had their effect on youngsters; a study conducted by Shoshani and Kor (2021) revealed a significant increase in psychological distress and mental health symptoms among school-aged children compared to the pre-pandemic baseline. However, to our best knowledge, no study has yet to examine the effect of the pandemic on risk behaviors of adolescents in Israel [8].

Risk behavior in adolescence is a socially unacceptable voluntary behavior with the potential for direct or indirect harm to a person, in which no precautions are taken, such as drinking alcohol and driving, drug use, tobacco smoking, frequent absences from school, delinquency, and more [9].

Specifically, the current study examines substance use of alcohol, cannabis, and cigarettes, which are the leading areas of public health concern for adolescents. Alcohol use in adolescence is considered a matter of public health concern due to its interrelation to additional risk behaviors, the tendency for alcohol use to persist into adulthood, and the impact it can have on the adolescent brain and health in general [9,10]. Alcohol use is the leading cause for cause-specific disability-adjusted life years (DALYs) for young people aged 10–24 years. Health risks behaviors experienced during adolescence often continue in adulthood, causing an increase in the rate of morbidity and mortality [11]. Specifically, smoking behavior is positively associated with many chronic diseases such as lung cancer, diabetes, and cardiovascular diseases, as well as affective disorders. Studies demonstrated that cannabis use is a potential cause for the development of psychotic disorders such as schizophrenia and may lead to permanent cognitive impairment. Additionally, cannabis use is associated with involvement in physical hazards and chronic diseases [9,10,11,12,13].

The Israeli law prohibits the sale of alcoholic beverages to minors and the consumption of any alcohol by youngsters is considered dangerous and harmful, though not prohibited. As for cannabis use, the Israeli law has imposed fines on adults caught consuming cannabis for personal use, but minors are exempt from the law and treated accordingly by the welfare authorities [14].

Protective factors reduce the probability of problematic behaviors through direct personal or social control and help the individual choose a healthier alternative path [15]. There are three classes of protective factors; personal protective factors include academic success, optimistic perception of life, and personal responsibility. The second is interpersonal protection factors, e.g., relationships with a significant person, such as parents, mentors, and teachers. The third is environmental protection factors, including an affiliation with the school and other social and community institutions, a strong family affinity, social support, and parental school involvement [15,16,17]. Studies point to the importance of family support in adolescents’ involvement in risky behaviors [17,18]. For example, the authoritarian parenting style was consistently associated with lower drug use rates among adolescents [19], while the uninvolved parenting style was associated with a tendency to use drugs among adolescents [20].

Recent studies demonstrate the effects of disasters and epidemics on children and adolescents, including anxiety, depression, psychological problems, and post-traumatic symptoms [21,22]. For example, an online convenience sample of 1442 e-cigarette users aged 13–20 years found that, in May 2020, 67% of users reported reduced use and 37% had quit entirely since the beginning of the pandemic [23]. Another study of 1054 Canadian teens aged 16–18 years found that significantly fewer teens reported binge drinking, cannabis use, and vaping after social distancing began, but alcohol and cannabis use frequency increased [24].

The primary purpose of this research was to investigate risky behaviors, such as tobacco smoking, alcohol drinking, and drug use, among adolescents in Israel during the COVID-19 pandemic and whether the pandemic influenced the prevalence of these risky behaviors. Furthermore, this study examined whether there are links between different risk behaviors during COVID-19 among adolescents in Israel and which characteristics can predict risk behaviors in adolescents.

## 2. Materials and Methods

### 2.1. Design

This was a cross-sectional design.

### 2.2. Participants and Procedure

The study participants consisted of 1020 Israeli youth aged 15 to 18 (57.3% girls, 42.7% boys). Data were collected between the first and second lockdowns in Israel (April 2020 to September 2020) through iPanel (https://www.ipanel.co.il/en/, accessed on 17 May 2021). iPanel is an online sampling service that allows fast responses, striving for a representative sample based on the population’s sociodemographic characteristics, such as age, gender, and health status. This 100,000-member panel is the largest panel survey in Israel and holds high-quality research codes from the European Society for Opinion and Marketing Research (ESOMAR) [25,26,27].

An introductory e-mail was sent to 2187 potential candidates via the iPanel database system explaining the research objectives in detail and their rights, such as the right to withdraw at any time from the research. The survey directions emphasized anonymity and confidentiality. Data collection resulted in 1020 full questionnaires, representing a response rate of 46.6%.

### 2.3. Measurements

The present study’s measurements were adopted from the Health Behavior in School-Aged Children (HBSC) organization’s protocol. The HBSC is a school-based survey of adolescent health behaviors and psychosocial determinants carried out among representative samples of school aged children every four years in more than 50 countries, using an international standardized methodological protocol [28,29,30] involving standardized procedures for the sampling and translation of items [28].

#### 2.3.1. Outcome Measures

Sociodemographic characteristics: The HBSC includes items describing participants’ sociodemographic characteristics. Participants reported their self-identified sex (boy, girl), age, and socioeconomic status using the revised Family Affluence Scale (FAS) [28,29].

Family support: Two separate scales assessing family support were used. In the first scale, answers ranged from 1 to 5 and in the second scale from 1 to 7. For each scale, all items were summed in order to create a total family support score. Higher scores represent greater family support. The scale’s Cronbach’s α = 0.94. Since the Pearson correlation between the two scales presented strong correlations (r = 0.567) between the two scales, an average score was calculated [28,30].

Friends Support: Friends support was measured using the Multidimensional Scale of Perceived Social Support (MSPSS) [30,31]. Answers to each question were given on a 7-point Likert scale, ranging from ‘strongly disagree’ to ‘strongly agree’. Items’ scores were summed in order to create a total score. The scale’s Cronbach’s α = 0.92.

Emotional Health: The HBSC 8-item emotional health symptom scale symptom checklist was administered to each participant [32] and has been used in all previous HBSC surveys. The scale is flexible in that statistical analyses are meaningful both on the single-item and sum score level [33,34]. Accordingly, a total score was calculated, with a higher score representing lower emotional health [35].

Health during COVID-19 pandemic: Participants were asked to describe their health during the COVID-19 pandemic using the question: “How would you describe your health during the COVID-19 pandemic?”, with answers ranging from 1, not good, to 4, excellent.

Physical activity level: Physical activity level was measured using the question “How often over the past seven days have you been physically active for a total of at least 60 min per day?”. Answers were given on an 8-point scale (0 = none to 7 = daily). The measure has reasonable validity (r = 0.37) with five-day accelerometer data [36] and acceptable test-retest reliability when used as a dichotomous variable [37]. Moderate-to-vigorous physical activity was also assessed with the following item: “How many hours a week do you engage, in your free time, in physical activity that makes you get out of breath or sweat?”, with scores ranging from 1 (0 h) to 6 (7 h/week or more).

COVID-19 restrictions: Participants were asked whether they were asked not to engage in any of the following behaviors during the COVID-19 pandemic: social gatherings, public transportation, isolation due to exposure to someone who might be positive for COVID-19, isolation on account of exposure to someone positive to COVID-19, and a family member hospitalized on account of COVID-19. Answers were either yes (1) or no (0). A total score of the number of restrictions was calculated.

#### 2.3.2. Outcomes Assessing Risky Behavior

Four different risky behaviors were assessed: risky behavior during the COVID-19 pandemic, tobacco smoking habits, alcohol consumption (binge drinking), and cannabis use.

Risky behavior during COVID-19: Participants scored the prevalence of eight risky behaviors (cigarette smoking, electronic cigarette smoking, drinking alcohol, using tranquilizer pills, smoking cannabis, using other illegal drugs, using new psychoactive substances, and using Ritalin) as either 0 (not presenting the risky behavior) or 1 (presenting the risky behavior). A total score reflecting the number of risky behaviors was calculated. The total score was dichotomized to those not presenting any risky behavior and those presenting one or more risky behaviors.

Tobacco smoking habits: The HBSC questionnaire includes mandatory question items investigating tobacco consumption. In the current study, the question that examined the frequency of tobacco smoking habits was “How often have you smoked tobacco in the past 30 days?”, with response options ranging from 1 (never) to 30 or more (7). Answerers were further dichotomized into participants who never smoked and those who smoked at least once [38].

Alcohol consumption (binge drinking): The HBSC instrument was used to collect information on alcohol consumption [39]. Binge drinking was assessed via the question: “In the past 30 days, how many times have you drank five drinks of alcohol or more within a period of a few hours?” (1—never; 2—not in the past month; 3—once; 4—twice; 5—3 times; 6—four times or more). In addition, a dichotomous variable was created to identify adolescents not involved in alcohol use (participants who answered the question with 1—never) [39].

Cannabis use: Cannabis use was measured based on a series of questions asking respondents how often they had used cannabis in their lifetime, the last year, and the previous month. Cannabis use experiences were coded: 0 = abstinence, 1 = experimental use (cannabis use 1–2 times in the last month or more, but less than ten times in their lifetime), 2 = regular use (three or more times in the past month and at least ten times in their lifetime). As the prevalence of experimental and regular users was small, a dichotomized variable was created (i.e., abstinence vs. experimental or regular users).

### 2.4. Data Analysis

Descriptive statistics (mean, standard deviation, range, and prevalence) as well as chi-square tests were used to describe participants’ main sociodemographic characteristics.

The prevalence of each risky behavior was dichotomized into those presenting and those not presenting the risky behavior. Chi-squared tests were used to examine differences in the prevalence of those presenting and not presenting the risky behavior. Associations between the four risky behaviors were also examined using Spearman rank correlation coefficients.

Changes in risky behavior during the pandemic, comprising eight different risky behaviors, were further analyzed. The prevalence of those who stopped, started, decreased, increased, or did not present any changes in their risky behavior during the pandemic were reported.

Differences in independent variables (i.e., sociodemographic characteristics, level of support, physical activity volume, corona restrictions, and health status) between participants presenting and not presenting risky behaviors were examined using Chi-squared (categorical variables) or independent *t*-tests (continuous variables). Variables significantly different between those presenting and not presenting the risky behaviors were entered into four separate binary logistic regression models in order to determine the extent to which the independent variables were predictive of the risky behaviors. In that respect, the dependent variables (i.e., the risky behaviors) were coded as 0, not presenting, and 1, presenting the risky behavior. All independent variables were checked for multicollinearity using the variance of the inflation factor. The criterion for inclusion in the model was an alpha level of 0.05. The data were analyzed with IBM SPSS statistics 19. In all statistical analyses, *p*-values lower than 0.05 were considered statistically significant.

### 2.5. Ethical Considerations

The study protocol was approved by the Ethics Committee of Ariel University, confirmation number: AU-HEA-RT-20210610. Participants were voluntarily recruited and informed of the research goals. Volunteers signed an informed consent form before answering the questionnaire and were assured that they had the right to withdraw from the research at any time, that their answers would be kept confidential, and that the questionnaires would be analyzed anonymously.

## 3. Results

### 3.1. Study Participants’ Characteristics

This study included 1020 youth aged 15 to 18 years old (mean age = 16.73 ± 0.99 years; 57.3% females). Participants were recruited from throughout the country; however, most study participants were from the center of the country (32.2% of the sample). For additional information, refer to Table 1.

### 3.2. Risky Behavior Characteristics

The most prevalent risky behavior in the sample was binge drinking, with 33.8% of the sample reporting experiencing binge drinking in the past month. The next most prevalent risky behavior was tobacco smoking in the last 30 days, with approximately 10% of the sample reporting this risky behavior (Table 2).

In addition, the four measures of risky behavior were significantly correlated (r range: 0.32 to 0.47; *p* < 0.05; Table 3).

In addition, the four measures of risky behavior were statistically significantly correlated (*p* < 0.0001). The highest correlation observed was between tobacco smoking in the last 30 days and total risky behavior during the pandemic (r = 0.47, *p* < 0.0001), and the lowest was between cannabis use in the last 30 days and total risky behavior during the pandemic (r = 0.32, *p* < 0.0001). For additional information, refer to Table 3.

The measure of risky behaviors during the COVID-19 pandemic comprises eight independent behaviors. The mean number of risky behaviors per participant was small (0.82 ± 1.29) and ranged between 0 and 8 (Figure 1).

Most of the participants did not change the frequency of their risk behavior during the pandemic (55.2%). However, more than 20% of the sample started to or increased their frequency of smoking cigarettes (20.7%), smoking electronic cigarettes (27.4%), anti-anxiety medications (31.4%), and smoking cannabis (30.6%). In addition, 14.4% of the participants started or increased the frequency of drinking alcohol (binge drinking), On the other hand, 20% decreased and 26.7% stopped using other illegal drugs during the pandemic, 15.9% reduced alcohol consumption, and 18.9% reduced cigarette smoking (Table 4).

### 3.3. Factors Related to and Predicting Risky Behavior

All independent variables, except for friends’ support, physical activity volume, and COVID-19 restrictions, significantly differed between participants engaging vs. those not engaging in risky behaviors. For example, in all risky behaviors assessed, the mean family support among those not presenting the risky behavior was significantly higher than those presenting risky behavior. Similarly, in all risky behaviors, as well as in total risky behavior during the pandemic score, those with higher emotional health scores (less favorable emotional health status) presented greater risky behaviors than those with lower scores (more favorable emotional health status; Table 5).

The regression model showed that tobacco smoking in the last 30 days was related to age (higher age predicting tobacco smoking) and emotional health (higher emotional distress predicting tobacco smoking; χ^2^ = 30.16, *p* < 0.0001). Similarly, older age and greater emotional distress predicted binge drinking. Male sex was also a significant predictor of binge drinking (Chi-squared = 40.98, *p* < 0.0001). In contrast, cannabis use was only predicted by family support, with greater family support predicting abstinence (Chi-squared = 14.47, *p* = 0.01). Total risky behavior during the pandemic was predicted by older age, emotional distress, and low family support (Chi-squared = 32.88, *p* < 0.0001). Health during the pandemic and socioeconomic status were not significant predictors of any of the risky behaviors evaluated (Table 6).

## 4. Discussion

The Coronavirus pandemic (COVID-19) has profoundly impacted adolescents’ lives [11,14]. This study examined the factors influencing risky behaviors (tobacco smoking, alcohol drinking, and drug use) among adolescents in Israel two months after the first lockdown in 2020. In addition, we examined the possible links between different risk behaviors, in the prevalence of risk behaviors, and which factors could predict risk behaviors in adolescents during COVID-19.

Our findings show that all risky behaviors during the pandemic were moderately correlated. These results are consistent with many other studies showing associations between risky behaviors such as illicit drugs, marijuana, alcohol, and e-cigarette use [40,41,42,43,44]. However, the analysis of risk behavior prevalence during the pandemic yielded mixed results. Approximately 50% of those engaged in risky behaviors did not change their behaviors during the pandemic. Alcohol consumption was somewhat decreased, with 30% reporting cessation or reduction of alcohol consumption, whereas 15% reported starting or increasing consumption. The observed decline may be due to the imposed lockdown and restrictions limiting the adolescents’ access to alcohol. Decreased Ritalin consumption during the pandemic (55% reported discontinuation or reduction) may be attributed to the forced transition to remote learning. In contrast, there was an increase in anti-anxiety drug use (31% started or increased use, compared with less than 17% who stopped or reduced), similarly to other studies reporting an increase in anti-anxiety drug use among adults during COVID-19. Moreover, international and non-college data demonstrate shifts in drinking behaviors related to the COVID-19 pandemic [45,46].

Analysis using a regression model revealed that all collected independent variables, except for friends’ support, physical activity volume, and coronavirus restrictions, significantly differed between participants engaging vs. not engaging in risky behaviors. In all risky behaviors assessed, the mean family support was significantly higher among those not presenting the risky behavior than those presenting it. This result is in line with multiple studies that point to the importance of parental support in preventing risky behaviors, such as reducing adolescent substance use [47,48]. Indeed, more involving parents are more successful in protecting their adolescents from problematic drug use [49]. Furthermore, setting clear rules and boundaries, open communication, and a good parent–child relationship are all associated with reduced use of illicit substances among adolescents [50]. In addition, in our study, emotional health was found to predict all risky behaviors, with the chance of engaging in risky behaviors increasing with increased emotional distress. Being younger and female also reduced the likelihood of engaging in risky behaviors, consistent with previous studies [51,52,53].

Interestingly, the present study found that socioeconomic status was positively correlated with risky behaviors during the pandemic. This may be because upper-class adolescents had more resources and access to illegal substances during the quarantine compared to lower-socioeconomic adolescents. In addition, they may have enjoyed greater personal space in their homes, allowing them to continue engaging in risky behaviors without being caught. Indeed, Hanson and Chen [54] found a link between low socioeconomic status and cigarette smoking, but no clear alcohol consumption or marijuana use pattern. Future research should examine how resource availability affects different kinds of risky behaviors during routine and in times of crisis.

Additionally, peer support and physical activity did not correlate with any risky behaviors. While being a protective factor, peer support may also encourage risk-taking actions, and their attitudes towards experimentation with substance use may be positive [55]. The opposite potential effect of peer support may explain its lack of contribution to predicting risky behaviors in the present study. Yet another possible explanation is that the COVID-19 restrictions had impaired social connections, thus lessening their protective essence [56,57,58]. Although previous studies have linked physical activity with adolescent health risk behaviors [59,60], the present study did not find evidence for such association, possibly due to an overall decrease in physical activity owing to COVID-19, as found in a recent report on young Spanish adults [61].

One of the strengths of the present study is the large response rate (46.6%) of participants that answered the online questionnaire (1020 out of 2187). However, a few limitations must be observed. First, the measures in this study were limited to adolescent self-report, without further testing. Therefore, several tools are needed to increase data reliability and prevent biases, such as a personal interview. Second, the sample was not probabilistic and included a group of adolescents looking to participate in this type of online research (register on the site); therefore, is not necessarily representative. It is crucial to make a representative sample of all adolescents across Israel to increase data reliability. Third, data were collected between the first lockdown in Israel (March–April 2020) and the second lockdown (September 2020). Thus, it should be taken into consideration that the results and the implications for adolescents’ use of substances may have been different if given later during the pandemic. Hence, it is recommended to conduct research sampled at several points in time to find a systematic, longitudinal research set-up.

## 5. Conclusions

Our findings suggest that family support is one of the most influential factors in preventing risky behavior during the pandemic, emphasizing the importance of family-based interventions with children and adolescents. It is necessary to allocate state budgets to deal with COVID-19 consequences, such as strengthening and encouraging adolescents who have not started or who have increased risky behaviors during this period and providing educational and therapeutic responses to adolescents who have used substances. In addition, targeted and tailored intervention programs for the adolescent population who drink alcohol are necessary. Future research should examine the characteristics of adolescents who started or increased the use of substances compared to those who stopped or reduced.

## Figures and Tables

**Figure 1 ijerph-19-03850-f001:**
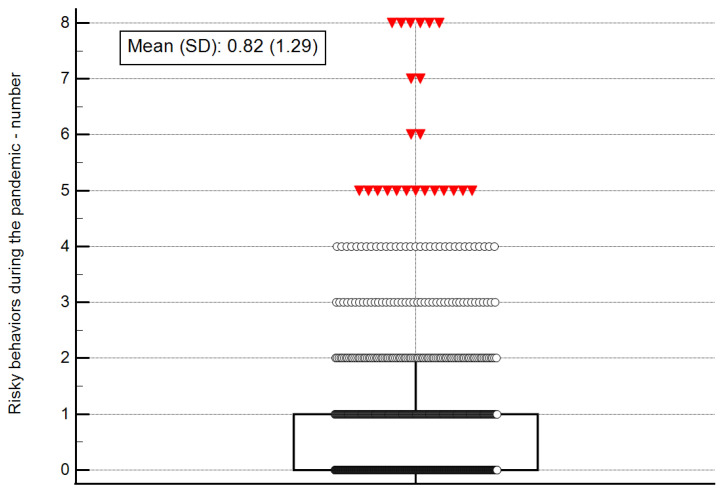
Number of risky behaviors during the pandemic. Note: SD, standard deviation. The central box represents the values from the lower to upper quartile (25 to 75 percentile); the vertical line extends from the minimum to the maximum value, excluding outside values, displayed as separate points. An outside value is defined as a value that is smaller than the lower quartile minus 1.5 times the interquartile range, or larger than the upper quartile plus 1.5 times the interquartile range.

**Table 1 ijerph-19-03850-t001:** Sociodemographic characteristics of study participants (*n* = 1020).

Variable		Mean (SD) [Range] OR *n* (%)	Chi-Squared (*p* Value)
Age, years: mean (SD) [range]		16.73 (0.99)	-
		[15.00–18.00]	
Sex: *n* (%)	Females	584 (57.3)	21.47 (<0.0001)
	Males	436 (42.7)	
Socio-economic status, score: mean (SD) [range] ^†^		13.80 (2.20)	-
		[7.00–19.00]	
Living area: *n* (%)	North of the country	188 (18.4) ^c,e^	130.48 (<0.0001)
	South and coastal plain	211 (20.7) ^c,e^	
	Center of the country	328 (32.2) ^a,b,d,e^	
	East of the country	193 (18.9) ^c,e^	
	Central coastal plain	100 (9.8) ^a,b,c,d^	

Notes: ^†^ The socioeconomic status scale scores range from 6 to 19, with higher scores representing higher status; ^a^, significantly different than “north of the country” (2-tailed, *p* < 0.05); ^b^, significantly different than “South and coastal plain” (2-tailed, *p* < 0.05); ^c^, significantly different than “Center of the country” (2-tailed, *p* < 0.05); ^d^, significantly different than “East of the country” (2-tailed, *p* < 0.05); ^e^, significantly different than “Central coastal plain” (2-tailed, *p* < 0.05); SD, standard deviation.

**Table 2 ijerph-19-03850-t002:** Risky behavior characteristics (*n* = 1020).

Risky Behavior			*n* (%)	Chi-Squared (*p*-Value)
Tobacco smoking	In last 30 days	Yes	93 (9.1)	681.91
		No	927 (90.9)	(<0.0001)
Drinking	Binge drinking	Yes	345 (33.8)	106.76
		No	675 (66.2)	(<0.0001)
Cannabis	Abstinence		952 (93.3)	766.13
	Experimental or regular		68 (6.7)	(<0.0001)
Risky behavior during the pandemic	No risky behavior		922 (90.4)	665.66
	One or more risk behavior		98 (9.6)	(<0.0001)

**Table 3 ijerph-19-03850-t003:** Associations between risk behaviors (Spearman rank correlation coefficients; *n* = 1020).

	Tobacco Smoking in Last 30 Days: r (*p* Value)	Cannabis Use in Last 30 Days: r (*p* Value)	Binge Drinking in Last 30 Days: r (*p*-Value)	Risky Behavior during the Pandemic: r (*p*-Value)
Tobacco smoking in last 30 days	-	0.33 (<0.0001)	0.40 (<0.0001)	0.47 (<0.0001)
Cannabis use in last 30 days	-	-	0.41 (<0.0001)	0.32 (<0.0001)
Binge drinking in last 30 days	-	-	-	0.41 (<0.0001)
Risky behavior during the pandemic	-	-	-	-

**Table 4 ijerph-19-03850-t004:** Changes in risky behavior during the pandemic among users.

Risky Behavior	Stopped Using during the Pandemic: *n* (%)	Started Using during the Pandemic: *n* (%)	Decreased the Amount of Use during the Pandemic: *n* (%)	Increased the Amount of Use during the Pandemic: *n* (%)	No Change during the Pandemic: *n* (%)
Smoking cigarettes (*n* = 106)	9 (8.5)	12 (11.3)	20 (18.9)	10 (9.4)	55 (51.9)
Smoking electronic cigarettes (*n* = 73)	10 (13.7)	18 (24.7)	7 (9.6)	2 (2.7)	36 (49.3)
Drinking alcohol (*n* = 359)	52 (14.5)	17 (4.7)	57 (15.9)	35 (9.7)	198 (55.20)
Anti-anxiety medications (*n* = 86)	8 (9.3)	11 (12.8)	8 (9.3)	16 (18.6)	43 (50.0)
Smoking cannabis (*n* = 62)	9 (14.5)	9 (14.5)	3 (4.8)	10 (16.1)	31 (50.0)
Other illegal drugs (*n* = 15)	4 (26.7)	0 (0.00)	3 (20.0)	1 (6.7)	7 (46.7)
New psychoactive substances (*n* = 16)	3 (18.8)	1 (6.2)	1 (6.2)	3 (18.8)	8 (50.0)
Ritalin (*n* = 124)	48 (38.7)	9 (7.3)	20 (16.1)	4 (3.2)	43 (34.7)

**Table 5 ijerph-19-03850-t005:** Differences between participants presenting and not presenting risky behavior.

Variables		Tobacco Smoking in the Last 30 Days		Binge Drinking		Cannabis Use		Risky Behavior during the Pandemic	
		Yes (*n* = 93)	No (*n* = 927)	Yes (*n* = 345)	No (*n* = 675)	Abstinence (*n* = 952)	Experimental or Regular (*n* = 68)	No Risky Behavior (*n* = 922)	One or More Risky Behaviors (*n* = 98)
Sex	Females, *n* (%)	46 (49.50)	538 (58.0)	177 (51.3)	407 (60.3)	546 (57.4)	38 (55.9)	533 (57.8)	51 (52.0)
	Males, *n* (%)	47 (50.5)	389 (42.0)	168 (48.7)	268 (39.7) ^†^	406 (42.6) ^†^	30 (44.1)	389 (42.2)	47 (48.0)
Age, years: mean (SD)		17.11 (0.88)	16.69 (0.99) *	16.90 (1.00)	16.64 (0.97) *	16.70 (0.98)	17.04 (0.94)	16.71 (0.99)	16.95 (0.91) *
Family support, total score: mean (SD)		17.32 (3.77)	18.21 (3.77) *	17.69 (3.76)	18.35 (3.77) *	18.21 (3.76)	16.41 (4.18) *	18.25 (3.76)	16.94 (3.74) *
Socio-economic status, score: mean (SD)		13.55 (2.21)	13.82 (2.19)	13.98 (2.27)	13.70 (2.15)	13.80 (2.19)	13.88 (2.40)	13.79 (2.19)	13.85 (2.27) *
Friends support, score: mean (SD)		20.55 (5.50)	21.17 (5.13)	21.47 (5.29)	20.94 (5.10)	21.16 (5.11)	19.80 (6.33)	21.18 (5.08)	20.47 (5.90)
Physical activity volume, days/week: mean (SD)		1.75 (1.89)	1.76 (1.90)	1.77 (1.91)	1.75 (1.89)	1.77 (1.90)	1.36 (1.62)	1.77 (1.90)	1.62 (1.89)
Corona restrictions, number: mean (SD)		2.32 (1.30)	2.16 (1.06)	2.21 (1.15)	2.15 (1.05)	2.16 (1.08)	2.38 (1.29)	2.16 (1.08)	2.30 (1.19)
Health during the pandemic, score: mean (SD)		3.32 (0.75)	3.50 (0.67) *	3.46 (0.69)	3.49 (0.68)	3.49 (0.68)	3.24 (0.77) *	3.50 (0.66)	3.29 (0.80) *
Emotional health, score: mean (SD)		21.37 (7.81)	18.84 (7.05) *	19.86 (7.29)	18.67 (7.06) *	18.91 (7.15)	21.52 (6.84) *	18.74 (7.09)	22.20 (7.10) *

Notes: ^†^ Significant differences in prevalence between males and females (2-tailed; *p* < 0.05); * significant between-group differences (2-tailed; *p* < 0.05); SD, standard deviation.

**Table 6 ijerph-19-03850-t006:** Summary of multiple binary logistic regression analysis for predicting risky behavior.

Dependent Variable	Predictors	Coefficient	Standard Error	Odds Ratio	Wald	95% CI	*p*-Value
Tobacco smoking in the last 30 days	Constant	−10.21	2.29		19.77		<0.0001
	Age, years	0.47	0.12	1.61	15.85	1.27–2.04	0.0001
	Family support, score	−0.03	0.02	0.96	1.40	0.91–1.02	0.23
	Health during the pandemic, score	−0.12	0.16	0.88	0.57	0.63–1.21	0.44
	Emotional health, score	0.04	0.01	1.04	6.46	1.00–1.07	0.01
	Model summary		Chi-squared = 30.16, *p* < 0.001, Nagelkerke R^2^ = 0.07.				
Binge drinking	Constant	−5.34	1.25		18.10		<0.0001
	Sex (reference—males)	−0.51	0.14	0.59	13.44	0.45–0.78	0.0002
	Age, years	0.29	0.06	1.34	18.28	1.17–1.54	<0.0001
	Family support, total score	−0.03	0.01	0.96	3.46	0.93–1.00	0.06
	Emotional health, score	0.03	0.01	1.03	9.03	1.01–1.05	0.002
	Model summary		Chi-squared = 40.98, *p* < 0.001, Nagelkerke R^2^ = 0.06.				
Cannabis use	Constant	−0.09	1.13		0.76		0.04
	Family support, total score	−0.09	0.03	0.91	5.97	0.84–0.98	0.01
	Health during the pandemic, score	−0.25	0.20	0.77	1.55	0.51–1.15	0.21
	Emotional health, score	0.02	0.02	1.02	1.39	0.98–1.07	0.23
	Model summary		Chi-squared = 14.47, *p* = 0.01, Nagelkerke R^2^ = 0.05.				
Risky behavior during the pandemic	Constant	−7.53	2.27		10.98		0.0009
	Age, years	0.29	0.11	1.34	6.79	1.07–1.67	0.0009
	Family support, total score	−0.06	0.02	0.94	4.43	0.88–0.99	0.03
	Socioeconomic status, score	0.04	0.05	1.04	0.82	0.94–1.15	0.36
	Health during the pandemic, score	−0.11	0.15	0.89	0.49	0.65–1.22	0.48
	Emotional health, score	0.05	0.01	1.05	11.48	1.02–1.09	0.0007
	Model summary		Chi-squared = 32.88, *p* < 0.0001, Nagelkerke R^2^ = 0.07.				

Note: Only variables that significantly differed between participants with/without risky behaviors were entered to the model; CI, confidence interval.

## Data Availability

Not applicable.

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
