# Peer review of "Risk Behaviors, Family Support, and Emotional Health among Adolescents during the COVID-19 Pandemic in Israel"

_ijerph, 2022, doi:10.3390/ijerph19073850_

Round 1

Reviewer 1 Report

The article is well written and timely.  The methodology sections could be improved by:

(1) A clearer explanation of the study participants.  The narrative as written is confusing.  For example, it is stated "The study participants consisted of 1,020 Israeli youth aged 15-18.  Questionnaires were handed out to students between the first lockdown which ended in April,2020  and the second lockdown in September 2020.    What students were these?  Were they from the iPanel sample?  If so, how were they identified and given a questionnaire?  Was the same group that received the introductory e-mail?  I really had trouble following this.

(2) Was the 46.6% response rate for completed, submitted surveys?  

(3) Under Outcome measures - Sociodemographic characters  HBSC is mentioned and not defined nor defined earlier in the manuscript.  Please spell out HBSC the first time stated with the HBSC in () then use HBSC after that.

(4) How this survey came together is not clear.  Questions are asked about family support, friends support, etc...   Please describe how the survey was compiled, did a panel of experts review it? Was it field tested and revised? What was the validity and/or reliability?  What was the readability level?  How many questions were in the questionnaire?

(5) Why was the rationale for choosing to assess the four different risky behaviors?  And to better contextualize this, it would good to provide the reader with some information on - with respect to Israel -  legal age for smoking and drinking alcohol, legal status of cannabis, etc..   

Author Response

Dear Reviewer,

Your insightful remarks are much appreciated.

Please see our responses below.

Thank you.

The article is well written and timely.  The methodology sections could be improved by: (1) A clearer explanation of the study participants.  The narrative as written is confusing.  For example, it is stated "The study participants consisted of 1,020 Israeli youth aged 15-18.  Questionnaires were handed out to students between the first lockdown which ended in April,2020 and the second lockdown in September 2020.    What students were these? 

We have added the following information on lines 107-115

Were they from the iPanel sample?  If so, how were they identified and given a questionnaire?  Was the same group that received the introductory e-mail?  I really had trouble following this.

We have added the following information on lines 107-110

(2) Was the 46.6% response rate for completed, submitted surveys?

We have added the following information regarding the study population, founds on lines 117-119

(3) Under Outcome measures - Sociodemographic characters  HBSC is mentioned and not defined nor defined earlier in the manuscript.  Please spell out HBSC the first time stated with the HBSC in () then use HBSC after that.

Thank you for catching this error – we have spelled out HBSC the first time it is mentioned in the text.

(4) How this survey came together is not clear.  Questions are asked about family support, friends support, etc...   Please describe how the survey was compiled, did a panel of experts review it? Was it field-tested and revised? What was the validity and/or reliability?  What was the readability level?  How many questions were in the questionnaire?

We have included more details on lines 120-135

 (5) Why was the rationale for choosing to assess the four different risky behaviors?  And to better contextualize this, it would be good to provide the reader with some information on - with respect to Israel -  legal age for smoking and drinking alcohol, legal status of cannabis, etc.. 

We have included more details about it on lines: 58-76

Reviewer 2 Report

I don't understand why they're referring to the pandemic as an epidemic in the introduction.  There is no premorbid data on this type of population prior to COVID-19 presented.

How do I know COVID-19 increased the binge drinking?

It must be some data on binge drinking in a similar population and that should be explained.

The data is primarily presented in tables rather than some type of explanation to the reader,  to make it easier to follow the tables.  I'm having trouble understanding the conclusion and a source of it that authoritative parenting prevents the risky behaviors.

Where are the parenting skill tests? In the paper, I don't think the scale in general used is particularly hbsc rigorous, reliable or that well validated, but it is often used !

An age range from 15 to 18 should also have an average age!  and this type of needs caveats  as self reporting is about substance use disorder is unreliable  high risk  kids/teens  need urine drug screens.

Caveats could be The study could be replicated using other tests i.e.  adolescent millon , mmpi , mbti etc or other psycho metrics should be reviewed or considered for other papers !  or other studies of adolescence and other alcohol drinking scales !

Even validation of the drinking could be done by other parties blood alcohol levels these are just basic caveats as self-report is weak! The low participation rate suggest how hard it is for kids. 

To be honest I don't understand if there was more attempt to get additional witness reliability!  Overall I like the paper it's an important attempt to give parents information as well as family practitioners/ psychiatrists / infectious disease docs info on  an important topic.

Author Response

Dear Reviewer,

Your insightful remarks are much appreciated.

Please see our responses below.

Thank you.

I don't understand why they're referring to the pandemic as an epidemic in the introduction.  There is no premorbid data on this type of population prior to COVID-19 presented.

We have changed the word "epidemic" to "pandemic".

How do I know COVID-19 increased the binge drinking? It must be some data on binge drinking in a similar population and that should be explained.

We added more reference about Alcohol consumption on lines 331-338

Table 4 presents changes in risky behavior during the pandemic. The table shows that 14.4% of the participants started or increased the amount of alcohol drinking (binge drinking). This information was added to the results section.

I'm having trouble understanding the conclusion and a source of it that authoritative parenting prevents the risky behaviors.

We revised the conclusion on lines 345-346

Where are the parenting skill tests? In the paper, I don't think the scale in general used is particularly HBSC rigorous, reliable or that well validated, but it is often used.

In the introduction we briefly provide information on parenting style. However, we did not examine parental skills, only family support. Two separate scales assessing family support were used. Since Pearson correlation presented strong correlations (r = 0.567) between the two scales, an average score was calculated. Information on this scale appears in the outcome measures section. 

An age range from 15 to 18 should also have an average age  

Yes, the study's age range is 15 to 18 years old. The mean age is 16.73 + 0.99 years. The mean age and standard deviation is mentioned in Table 1 and in the text that describes Table 1 on line 234

57.3% females and this type of needs caveats  as self reporting is about substance use disorder is unreliable  high risk  kids/teens  need urine drug screens. Caveats could be The study could be replicated using other tests i.e.  adolescent millon , mmpi , mbti etc or other psycho metrics should be reviewed or considered for other papers !  or other studies of adolescence and other alcohol drinking scales !Even validation of the drinking could be done by other parties blood alcohol levels these are just basic caveats as self-report is weak! The low participation rate suggest how hard it is for kids. 

To be honest I don't understand if there was more attempt to get additional witness reliability!  

Reliability – the survey was also administered among 20 adolescents in order to check the scale's internal reliability and coherence.  Cronbach's alpha = 0.82.

Overall I like the paper it's an important attempt to give parents information as well as family practitioners/ psychiatrists / infectious disease docs info on  an important topic.

Thank you !

Reviewer 3 Report

This is a very well-done study in much needed field and context; thus, I have some minor suggestions for the authors to consider.

I would suggest to drop from the title “period” and add “pandemic in Israel”

As an international reader, I would like to know more about the local context, country specifics, youth population, major risk behaviors and issues affecting them, other similar studies, reports, programs; pandemic situation; why this context is unique and novel. Maybe is good to elaborate on the impact of the pandemic on these youth and what approach may be useful in such challenging times.

Thanks for the opportunity to be part of the editorial process of this paper and I wish the authors much success in their relevant work.

Author Response

Dear Reviewer,

Your insightful remarks are much appreciated.

Please see our responses below.

Thank you.

This is a very well-done study in much needed field and context; thus, I have some minor suggestions for the authors to consider.

Thank you so much

I would suggest to drop from the title “period” and add “pandemic in Israel”

We have made the suggested changes.

As an international reader, I would like to know more about the local context, country specifics, youth population, major risk behaviors and issues affecting them, other similar studies, reports, programs; pandemic situation; why this context is unique and novel. Maybe is good to elaborate on the impact of the pandemic on these youth and what approach may be useful in such challenging times.

We have added more information regarding the context specific to Israel on lines 46-54

Thanks for the opportunity to be part of the editorial process of this paper and I wish the authors much success in their relevant work.

Thank you very much!
